# Posture Assessment in Dentistry for Different Visual Aids Using 2D Markers

**DOI:** 10.3390/s21227717

**Published:** 2021-11-19

**Authors:** Alberto Pispero, Marco Marcon, Carlo Ghezzi, Domenico Massironi, Elena Maria Varoni, Stefano Tubaro, Giovanni Lodi

**Affiliations:** 1Azienda Ospedaliera Santi Paolo e Carlo, Unità Operativa Complessa Odontostomatologia II, Università degli Studi di Milano, Via Beldiletto 1/3, 20142 Milan, Italy; pispero.alberto@gmail.com (A.P.); elena.varoni@unimi.it (E.M.V.); giovanni.lodi@unimi.it (G.L.); 2Dipartimento di Elettronica, Informazione e Bioingegneria (DEIB), Politecnico di Milano, Piazza Leonardo da Vinci, 32, 20133 Milan, Italy; stefano.tubaro@polimi.it; 3Private Practice, Via G. Verdi 4, 20019 Settimo Milanese, Italy; segreteria@studioghezzi.info; 4Private Practice, Via L. Cadorna, 21, 20077 Melegnano, Italy; info@gandini-massironi.it

**Keywords:** clinical studies/trials, computer vision for assistive technologies, health services research, diagnostic systems, clinical practice guidelines

## Abstract

Attention and awareness towards musculoskeletal disorders (MSDs) in the dental profession has increased considerably in the last few years. From recent literature reviews, it appears that the prevalence of MSDs in dentists concerns between 64 and 93%. In our clinical trial, we have assessed the dentist posture during the extraction of 90 third lower molars depending on whether the operator performs the intervention by the use of the operating microscope, surgical loupes, or with the naked eye. In particular, we analyzed the evolution of the body posture during different interventions evaluating the impact of visual aids with respect to naked eye interventions. The presented posture assessment approach is based on 3D acquisitions of the upper body, based on planar markers, which allows us to discriminate spatial displacements up to 2 mm in translation and 1 degree in rotation. We found a significant reduction of neck bending in interventions using visual aids, in particular for those performed with the microscope. We further investigated the impact of different postures on MSD risk using a widely adopted evaluation tool for ergonomic investigations of workplaces, named (RULA) Rapid Upper Limb Assessment. The analysis performed in this clinical trial is based on a 3D marker tracker that is able to follow a surgeon’s upper limbs during interventions. The method highlighted pros and cons of different approaches.

## 1. Introduction

Improving dental care shifted the focus to conservative treatment, making necessary the use of minimally invasive techniques. Nowadays, the goal of treatment is to preserve dental tissues with minimally invasive prosthetics, conservatives, and endodontics therapies, and to reconstruct periodontal soft and hard tissues performing microsurgery treatments in order to get the best aesthetic results with the least pain and complications.

In dentistry, we have two kinds of magnification systems: loupes and operating microscopes. The first is a wearable device, with a Galilean or Keplerian magnification system widespread among dentists. Furthermore, it is possible to have a coaxial illumination that improves visual acuity in the oral cavity [1]. A recent review [2] on visual acuity in dentistry, studied the ability to perceive fine details of an object by comparing all magnification systems with the naked eye. In that study, Galilean or Keplerian loupes and the surgical microscope were considered in two groups of practitioners divided depending on age: less than 40 years or older. The article [2] points out how age can affect visual acuity and how the type of magnification device can improve it.

Based on the results of this research and other studies, [3] the microscope seems to be the better choice to get the best results in terms of visual acuity for a dentist, independent of his/her age.

Although the use of the microscope is increasing and the benefits of optical magnification are widely recognized, its general use remains limited in dental practice [4]. The main advantages of the microscope are not only related to high magnification, but also in changing the working distance and depth of field, keeping an upright position, and being able to quickly generate a complete operative iconography before, during and after treatments. All these advantages can find application not only in endodontics but in many other areas as well.

According to the World Health Organization (WHO), Work-related Musculoskeletal Disorders (WMSD) describe a wide range of inflammatory and degenerative diseases and disorders that result in pain and functional impairment. WMSDs are identified as injuries that can occur from a single event, or cumulative traumas, including any complaint, from slight transitory discomforts to irreversible and incapacitating injuries.

Attention and awareness towards WMSDs in the dental profession has increased considerably in recent years. Even after the evolution to seated four-handed dentistry and ergonomic equipment, studies found back, neck, shoulder, or arm pain present in up to 81% of dental operators [5,6,7]. This pain can be attributed to numerous risk factors, including prolonged static postures (PSPs), repetitive movements, suboptimal lighting, poor positioning, genetic predisposition, mental stress, physical conditioning, and age.

The injury prevention guidelines emphasize an upright posture for the dental clinician, offering stabilization of the trunk, easy reach of adopted equipment, close accessibility to the oral cavity, and the ability to change position frequently to improve access [8]. Research shows that maintaining the low back curve—the lumbar lordosis—when sitting can reduce or prevent low back pain; in particular, tilting the seat angle slightly forward 5 to 15 degrees can increase the low back curve.

Furthermore, the use of magnification aids and indirect vision systems allows operators to maintain healthier postures keeping a safe work distance and also preventing neck pain [9].

In order to assess, from a quantitative viewpoint, the impact of different visual aids on the posture with respect to the naked eye interventions, we adopted an unintrusive and cost-effective approach to accurately track and monitor the posture of upper limbs during surgical procedures. By applying specific markers placed on the practitioner’s back and processing their displacement, we have been able to get an accurate estimation of the complete limb motion using a novel computer vision approach based on quaternion algebra.

Furthermore, in order to evaluate objectively the risk of WMSD we used a widely adopted approach named Rapid Upper Limb Assessment (RULA) [10,11,12,13]; a survey method developed for use in ergonomic investigations of workplaces where work related upper limb disorders are reported. RULA is a screening tool that assesses biomechanical and postural loading over the whole body, with particular attention on the neck, trunk, and upper limbs. Reliability studies have been conducted using RULA on groups of VDU (Visual Display Unit) users and sewing machine operators. A RULA assessment consists of a multiple-answer form whose scoring points out criticalities and generates an action list. Every upper body part RULA result suggests the level of intervention required to reduce the risks of injury due to physical loading on the operator [14]. Even if the RULA method is one of the most commonly adopted in industrial environments, its results are commonly based on the subjective evaluation of angles and postures performed by a trained investigator from a direct observation or a movie. On the contrary, in our approach every limb angle and posture persistence is objectively noted.

## 2. Materials and Methods

The study is a randomized controlled three arms clinical trial. It is in collaboration with Politecnico di Milano, Dipartimento di Elettronica, Informazione e Bioingegneria (DEIB).

Between March 2017 and May 2018, at the Azienda Ospedaliera Santi Paolo e Carlo, Unità Operativa Complessa Odontostomatologia II in Milan (Italy), 90 extractions of lower third on 65 patients were carried out. All those patients who needed to extract both lower molars were treated in two separate sessions, and the second intervention was performed only after complete healing.

Patients were asked to sign a specific informed consent form to take part in the research protocol. The patients were enrolled at the aforementioned dental clinic, which is a public structure affiliated to the National Health System where people that meet the essential healthcare levels are eligible for free medical care.

A randomization in three groups with split in 38th and 48th tooth (respectively left and right lower molars) was performed using the website software Quickcalcs (Graphpad software [15]):Operating microscope.Surgical Galilean loupes with coaxial illumination.Naked eye.

Per each group, 15 extractions were carried out on the left side and 15 on the right one.

The intervention was performed under local anesthesia and by the use of a microscope (OPMI Movena S7 from Carl ZEISS S.p.A., Milano, Italy), surgical loupes (EyeMag from Carl ZEISS S.p.A., Milano, Italy), or no magnifying system at all (see Figure 1). The surgeon did not know in advance with which method the tooth extraction will take place, in order to avoid any pre-orientation. A few minutes before the intervention, a student opened the opaque envelope of randomization.

During this session, the following data was collected for the study:Demographic data (baseline)Medical and dental historyParameters related to the lower third molarRadiographic testsMarker operator positionPre- and post-operative photographs

The procedure was performed by a single operator: an expert oral surgeon assisted by a dental student. The operator is also an expert user of magnifying systems in oral surgery.

We considered the interventions performed on the left side and the right side in the same way. The operator sat on the same side of the extracted teeth.

The primary outcome of the study was to investigate the posture evolution and persistence of fatiguing positions of the dentist during interventions due to the chairside dentist work. To obtain the values needed for the posture assessment, we proposed a specific posture analysis tool using fiducial markers. In this case, no load transfer or wide and rapid motions were involved, and the main issues were related to static postures. In order to evaluate the dentist posture accurately and objectively, we applied a set of markers on the back of a tight T-shirt worn by the dentist during the whole operation; two further markers were applied on the surgical cap to estimate head position (see Figure 2).

The acquisitions were performed using a 5 MP camera at 2 fps. The camera was a Genie Nano M2420 CMOS (from Teldyne DALSA, Ontario, Canada), it is a global shutter, monochromatic camera using the GigE interface. The camera is able to reach more than 20 fps, however, since we skipped any lossy compression and related video codecs to avoid a possible blurry effect on the markers, we saved each single frame (around 2.5 MB) in PNG format with the associated time-stamp. The whole set of all the interventions (almost 65 h) takes more than 1.2 TB.

The 2 fps were largely sufficient to properly track slowly varying posture during surgeon activity.

The markers are based on the ArUco Library [16,17], which allows a robust and effective estimation of the absolute rototranslation of each marker with respect to the camera of each visible marker in the 3D space. Thanks to this approach it is possible to estimate accurately, per each frame, the posture of the upper limbs (Figure 3).

In literature and on the market, there are many off the shelf Mocap (Motion Capture) solutions, however, most of them are costly and cumbersome, finalized to an accurate 3D tracking of every limb, even in case of occlusions and rapid motions. Furthermore, most of them require the user to wear uncomfortable dark suits with high contrast markers. In our case we do not have large displacements of the user and the movements during the intervention must be as natural as possible without stretch from the suits.

The printed markers are pasted on thin corrugated plastic squares in order to preserve their flatness; the tiles are then linked on the t-shirt and the surgical cap using Velcro straps. The t-shirt can be put on and taken off without removing the markers; furthermore, they do not require an accurate positioning or alignment since, as explained below, we are just using their averaged differential positions and rotations.

We positioned the markers in order to focus our analysis on the following parameters:The neck position with respect to the trunk;The trunk orientation with respect to the vertical axis;The twist and bending of the neck and back;The overall static position persistence of the neck and the back.

In Figure 4, we overlapped the different positions of the ArUco markers during three different types of surgical procedures. As can be seen on the left, where we reported the ArUco positions for the naked eye, the posture changed deeply during the operation, with a higher bending of upper limbs in respect to the operation with the medical loupes (Figure 4—center) and, even more, in respect to the operation with the surgical microscope (Figure 4—right).

The positions and rotations of other joints, such as elbows and wrists, and limbs, like forearms and hands, are relevant and considered in the generic RULA approach; however, in our application their tracking would request two further cameras in order to acquire both the left and right side of the surgeon. The whole architecture would become more expensive and cumbersome; furthermore, the streams from each camera must be synchronized and occlusions due to the dental unit must be handled. We checked separately, on a few interventions with different visual aids, the variations on the forearms using a couple of self-powered wearable accelerometers and we did not find any significant variation between the three configurations (naked eye and visual aids).

We can then conclude that the proposed approach represents a cheap and accurate system to monitor the head, the neck, and the back of an operator, i.e., it is particularly suited to monitor the spine and possible related diseases. If we are also interested in a detailed tracking of the arms, we can add further cameras that would have to be synchronized and calibrated with the main one: the Extrinsics parameters could be obtained during the calibration phase, framing the checkerboard from each camera and then extracting the relative rotations matrices [18]. Another possible approach that is not particularly intrusive for forearm motion and angles estimation can be based on 7 DoF IMUs (7 Degrees of Freedom Wearable Inertial Measurement Unit), see, e.g., [19].

### 2.1. Intervention Procedure Description

After performing the local anesthesia, a few minutes were devoted to checking the camera and setting up the ideal (neutral) starting posture. When everything was ready, a countdown started and the recording began.

The first surgical step recorded was the *incision*, as we decided not to include the anesthesia step in this study since it is not related to the static posture during interventions.

We tracked the postural evolution of the dentist’s backbone, neck, and head during the whole operation, in particular, for each group (naked eye, surgical loupes, and microscope) and we collected data from neck and trunk forward bending, side bending and twisting.

The rotation angles of every marker axis were extracted and stored for every frame of the recorder video; the detailed processing procedure, based on quaternions, is detailed in the data analysis section.

Left and right side interventions were evaluated in the same way. As will be detailed in the results, we did not notice any significant variation in postures between left and right side interventions; therefore, left side or right side bending were considered in the same way just evaluating the side bending in absolute value. This consideration also applies to neck and trunk yaw and roll.

RULA worksheet (see Figure 5)) was used to analyze data and obtain the WMSDs risk score. In the form, there is a set of boxes to be filled in order to assess the final WMSDs risk. If we look at the dentist posture during the extraction of third lower molar for different magnifying systems, the only variables that change are neck rotation and trunk inclination. We then assigned the same score to all interventions in the Table A and Table B boxes of Figure 5. The value that changes, according to different magnifying systems, is represented with a “X” in Figure 5 and, introducing its value, provides the final RULA score.

### 2.2. Marker Tracking and Analysis

All the angles and displacements of the markers were measured with respect to the “reference posture” acquired at the very beginning of each intervention, where the surgeon is asked to keep a neutral position.

Marker rotations and translations were also recorded in an absolute reference system assumed on a horizontal plane with the Z axis vertical on that plane—this allowed us to properly handle the role of gravity in the analyzed data.

From the OpenCV ArUco Module [20] for each ArUco marker and for the horizontal reference checkerboard, we get the rotation matrices and the translation vectors with respect to the camera reference frame.

The accuracy in estimating localization and rotation of each ArUco marker depends on many factors; among them, the size in pixels, the viewing angle, and the region contrast play a major role. In our application, the distance of the markers varies in a short range and the whole environment is well illuminated; some difficulties in accurate detection can arise from oblique rotations, however, according to [21,22,23] the rotation estimation error, in a similar environment, is within ±1 deg. However, as detailed below, in our application we estimate the average rotation of trunk and head from the average of multiple markers, further reducing the estimation error.

In postural analysis, in particular concerning the upper body, the relative translations between branches are rarely considered [14] since the most relevant parameters are the relative rotations of joints. In the performed analysis, we then focused on joint relative rotations that can be easily handled using the compact quaternion notation instead of rotation matrices. In the following section, we provide a brief overview of quaternions with the principal algebraic operations that we used to analyze the surgeon’s posture.

#### 2.2.1. Quaternions Definition

Quaternions [24] represent a compact form to deal with vectors rotations with a more intuitive and explicit representation with respect to 3 × 3 spatial rotation matrices. We use the notation:(1)$$q=s+v=s+xi+yj+zk={\left[\begin{array}{cccc} s & x & y & z \end{array}\right]}^{⊺}$$
where s is a scalar and v is a 3D vector whose direction represents the rotation axis. We will use unit quaternions, i.e., quaternions whose magnitude is equal to one unit so that it can be assumed belonging to a hypersphere with unit radius, in a 4D space: $q\in S^{3}$. In particular, we consider:(2)$$q=s+\lambda \hat{v}$$
where $\hat{v}$ is a unit vector ($\left|\hat{v}\right|=1$) and $s^{2}+\lambda^{2}=1$. The quaternion can be reformulated as:$$q=\cos\left(\theta /2\right)+\sin\left(\theta /2\right)\hat{v}$$
where $\hat{v}$ is the unit vector indicating the rotation axis and θ is the rotation angle.

The other kind of quaternions that we use are pure quaternions, i.e., s=0 in Equation (1): they represent common 3D vectors, $\vec{p}=0+\lambda \hat{v}$, but this formulation allows to easily apply rotations to quaternions using the quaternion product:$${\vec{p}}^{\prime }=q\vec{p}q^{-1}$$
where $\vec{p}$ is the original 3D vector embedded in a pure quaternion, ${\vec{p}}^{\prime }$ is still a pure quaternion and represents the rotated vector and $q^{-1}$, only for unit quaternions, is obtained changing the sign of the vector component of the original quaternion.
$$q=s+v\Leftrightarrow q^{-1}=s-v$$

Further details on quaternions algebra can be found, e.g., in [24].

#### 2.2.2. Rotation Matrices Estimation from ArUco Markers

From each rotation matrix obtained for each ArUco marker we build the corresponding quaternion; the rotation axis can be obtained from the rotation matrix applying the eigenvectors-eigenvalues decomposition. The rotation axis will be the eigenvector associated to the eigenvalue 1 since the vectors along the rotation axis do not rotate when the rotation is performed:$$R_{3\times 3}\hat{v}=1\cdot \hat{v}$$

The rotation angle can be estimated from the trace of the rotation matrix or from the sum of its eigenvalues.
$$\mathrm{tr}\left(R_{3\times 3}\right)=1+2\cos\left(\theta \right)$$

We recall that the eigenvalues for a rotation matrix are $1,e^{i\theta },e^{-i\theta }$.

To disambiguate the rotation direction, since $\cos\left(\theta \right)=\cos\left(-\theta \right)$, we follow the approach proposed in [25]: according to the Rodrigues’ formula, for a generic vector u,
$$\begin{array}{l} R_{3\times 3}u=\left(1-\cos\theta \right)\hat{v}\left(\hat{v}\cdot u\right)+\cos\theta u+\sin\theta \hat{v}\times u=\\ =\left(1-\cos\theta \right)\left(\hat{v}{\hat{v}}^{⊺}\right)u+\cos\theta u+\sin\theta {\left[\hat{v}\right]}_{\times }u \end{array}$$
where ${\left[\hat{v}\right]}_{\times }=\left[\begin{array}{ccc} 0 & -v_{3} & v_{2}\\ v_{3} & 0 & -v_{1}\\ -v_{2} & v_{1} & 0 \end{array}\right]$ is the matrix formulation of the cross product.

We can then write that
(3)$$R_{3\times 3}=\left(1-\cos\theta \right)\left(\hat{v}{\hat{v}}^{⊺}\right)+\cos\theta I_{3\times 3}+\sin\theta {\left[\hat{v}\right]}_{\times }$$
where $I_{3\times 3}$ is the identity matrix.

Since
$$\hat{v}\left(\hat{v}\cdot u\right)=u+\hat{v}\times \left(\hat{v}\times u\right)\Rightarrow \left(\hat{v}{\hat{v}}^{⊺}\right)u=u+{\left[\hat{v}\right]}_{\times }{\left[\hat{v}\right]}_{\times }u$$
then $\hat{v}{\hat{v}}^{⊺}=I_{3\times 3}+{\left[\hat{v}\right]}_{\times }{\left[\hat{v}\right]}_{\times }$, so that we can rewrite Equation (3) as:$$\begin{array}{l} R_{3\times 3}=\left(1-\cos\theta \right)\left(I_{3\times 3}+{\left[\hat{v}\right]}_{\times }{\left[\hat{v}\right]}_{\times }\right)+\cos\theta I_{3\times 3}+\sin\theta {\left[\hat{v}\right]}_{\times }=\\ =I_{3\times 3}+\sin\theta {\left[\hat{v}\right]}_{\times }+\left(1-\cos\theta \right){\left[\hat{v}\right]}_{\times }{\left[\hat{v}\right]}_{\times } \end{array}$$

Pre-multiplying the equation above by ${\left[\hat{v}\right]}_{\times }$ we get:$${\left[\hat{v}\right]}_{\times }R_{3\times 3}={\left[\hat{v}\right]}_{\times }+\sin\theta {\left[\hat{v}\right]}_{\times }{\left[\hat{v}\right]}_{\times }+\left(1-\cos\theta \right){\left[\hat{v}\right]}_{\times }{\left[\hat{v}\right]}_{\times }{\left[\hat{v}\right]}_{\times }$$
and considering that
$${\left[\hat{v}\right]}_{\times }{\left[\hat{v}\right]}_{\times }=\hat{v}{\hat{v}}^{⊺}-I_{3\times 3}$$
$${\left[\hat{v}\right]}_{\times }\hat{v}=0$$
$${\left[\hat{v}\right]}_{\times }{\left[\hat{v}\right]}_{\times }{\left[\hat{v}\right]}_{\times }=-{\left[\hat{v}\right]}_{\times }$$
we get:$$\begin{array}{l} {\left[\hat{v}\right]}_{\times }R_{3\times 3}={\left[\hat{v}\right]}_{\times }+\sin\theta {\left[\hat{v}\right]}_{\times }{\left[\hat{v}\right]}_{\times }+\left(1-\cos\theta \right){\left[\hat{v}\right]}_{\times }{\left[\hat{v}\right]}_{\times }{\left[\hat{v}\right]}_{\times }=\\ ={\left[\hat{v}\right]}_{\times }+\sin\theta {\left[\hat{v}\right]}_{\times }{\left[\hat{v}\right]}_{\times }-\left(1-\cos\theta \right){\left[\hat{v}\right]}_{\times }=\cos\theta {\left[\hat{v}\right]}_{\times }+\sin\theta {\left[\hat{v}\right]}_{\times }{\left[\hat{v}\right]}_{\times } \end{array}$$

Since $\mathrm{Tr}\left({\left[\hat{v}\right]}_{\times }\right)=0$ and $\mathrm{Tr}\left({\left[\hat{v}\right]}_{\times }{\left[\hat{v}\right]}_{\times }\right)=-2$ for unit vectors we get:$$\mathrm{Tr}\left({\left[\hat{v}\right]}_{\times }R_{3\times 3}\right)=-2\sin\theta$$
and the rotation in the range of 0≤θ≤π is clockwise or counter-clockwise according to the positive or negative value of $\mathrm{sign}\left(-\mathrm{Tr}\left({\left[\hat{v}\right]}_{\times }R_{3\times 3}\right)\right)$, respectively.

#### 2.2.3. Quaternions Reformulation of Rotation Matrices

According to the aforementioned analysis of the rotation matrices, we obtained all the elements to associate a quaternion to each ArUco marker. Thanks to their compact formulation, quaternions represent an effective and intuitive tool to analyze the markers’ evolution in time and to estimate limb motion.

Further details on quaternions computations from rotation matrices can be found in [26].

To give a global reference to the whole framework with respect to the vertical direction we acquired a horizontal reference target named “ground” as depicted in Figure 6.

In the following we will call (see Figure 7): $q_{g}$ the quaternion representing the rotation from the camera to the horizontal floor represented by the checkerboard; $q_{l}\left(t\right)$, with 1≤l≤n representing the *l*-th marker (in our acquisitions we used 18 markers) and t represents the considered frame; t=0 is associated to the reference posture acquired at the very beginning of each intervention.

In order to refer all the markers to the horizontal plane we used the “ground” reference system described above and we apply the rotation from checkerboard to camera (the inverse rotation of the $q_{g}$ quaternion in Figure 7) and, then, from camera to each marker.

We call “referenced vectors” the pure quaternions in the horizontal plane reference system:$${\vec{p}}_{l}\left(t\right)=q_{l}\left(t\right)q_{g}^{-1}\vec{p}q_{g}q_{l}^{-1}\left(t\right)=\left(q_{l}\left(t\right)q_{g}^{-1}\right)\vec{p}{\left(q_{l}\left(t\right)q_{g}^{-1}\right)}^{-1}$$

Relative rotations of a vector between two different frames $t_{1}$ and $t_{2}$ can be described combining quaternions of each frame:$${\tilde{q}}_{l}\left(t_{2},t_{1}\right)={\left(q_{l}\left(t_{1}\right)q_{g}^{-1}\right)}^{-1}\left(q_{l}\left(t_{2}\right)q_{g}^{-1}\right)=q_{g}q_{l}{\left(t_{1}\right)}^{-1}q_{l}\left(t_{2}\right)q_{g}^{-1}=q_{l}{\left(t_{1}\right)}^{-1}q_{l}\left(t_{2}\right)$$

Another useful property of quaternions is that they can be easily averaged from a set of rotations of different markers. This allows us, in our analysis, to find the mean value of relative rotations of different body parts. In particular, a common approach to find the average quaternion is based on the eigenvector decomposition, which allows us to directly solve two issues related to quaternion average; since quaternions provide a 2:1 mapping of the rotation group [27], two equal quaternions with opposite signs represent the same rotation: ${\vec{p}}^{\prime }=q\vec{p}q^{-1}=\left(-q\right)\vec{p}\left(-q^{-1}\right)$. Therefore, quaternions cannot be simply summed and averaged; furthermore, the resulting average quaternion has to belong to a glome: $\bar{q}\in S^{3}$. A common way to solve this problem, according to [28] is described below.

Given a weighted 4×4 correlation matrix M:(4)$$M=\sum_{i=1}^{n}w_{i}q_{i}q_{i}{}^{⊺}$$

Its eigenvector, associated to the largest eigenvalue, represents the average quaternion, i.e.,:$$\bar{q}=\arg\underset{q\in S^{3}}{\max}q^{⊺}Mq$$

In Equation (4) the weights can be simply $w_{i}=\frac{1}{n}$ or could consider the accuracy of quaternions estimated for different markers, i.e., for markers farther from the camera or with a more grazing position, the weight could be lower with respect to more frontal or close to the camera targets.

In our analysis we gave the same weight to every marker. The issue related to the quaternion sign ambiguity is avoided in Equation (4), since, even if $q_{i}$ changes its sign, the values of M do not change.

The last function that we used from quaternions algebra is quaternion decomposition along a specific axis. In particular, we adopted the Swing-Twist decomposition according to Huyghe’s method [29,30], where the twist angle represents the rotation along the desired axis while the swing is associated to a rotation on an axis orthogonal to the desired one. According to Huyghe, every quaternion can be written in terms of $q=q_{⊥}q_{\|}$ where $q_{\|}$ is the quaternion along the desired (twist) axis, while $q_{⊥}$ is the rotation along an orthogonal (swing) axis. The $q_{\|}$ unit quaternion can be easily obtained projecting the quaternion 3D vector component v (see Equation (1)) on the twist axis unit vector ${\hat{v}}_{\|}$ and normalizing the result to a unit quaternion:(5)$$q_{\|}=\frac{s+\left(v\cdot {\hat{v}}_{\|}\right){\hat{v}}_{\|}}{\sqrt{s^{2}+{\left(v\cdot {\hat{v}}_{\|}\right)}^{2}}}=\cos\left(\tau /2\right)+\sin\left(\tau /2\right){\hat{v}}_{\|}$$

We can then retrieve the rotation angle τ along the twist axis.

### 2.3. The Data Analysis

In order to process the data acquired tracking the ArUco markers we focused our analysis on the surgeon’s back and head since those regions are mostly exposed to WMSD and, according to the RULA form (Figure 5), the impact of different magnification devices is more evident.

We analyzed the back posture using from the 4th to the 18th ArUco markers (see Figure 2) and the head relative posture using from the 1st to 3rd markers. In particular, we considered the average trunk relative rotation defining the quaternion ${\tilde{\bar{q}}}_{t}$ which is obtained averaging the markers on the trunk at each frame and evaluating the relative rotation between two different instants of time. The other considered average relative quaternion is related to the head markers: ${\tilde{\bar{q}}}_{h}$; in Figure 8 it is depicted by the relative average quaternion for the trunk according to yaw (left), pitch (center) and roll (right) rotations. In the same way, in Figure 9, it is depicted by the neck-head relative average rotation ${\tilde{\bar{q}}}_{h}$ for yaw (left), pitch (center) and roll (right) rotations.

In the performed analysis, we considered the evolution of ${\tilde{\bar{q}}}_{t}$ and ${\tilde{\bar{q}}}_{h}$ during the different interventions with different magnification devices. In particular, we analyzed their mean value during interventions, their variance, and their persistence across frames (associated to long-term static postures). For each analyzed quaternion we decomposed it, using Equation (5), along different reference directions (see, e.g., Figure 8 and Figure 9) in order to assign the proper rotation angle to each field in the RULA form (Figure 5).

## 3. Patients Selection Criteria and Intervention Details

In the first appointment, the need for extraction of the third lower molar was verified.

All patients were treated in an outpatient setting. During the first visit, the clinician collected the patient’s remote and next medical history and carried out an accurate, objective examination of the oral mucous membranes and dental arches. The clinician also assessed if the patient met the inclusion criteria of the study:Presence of a third lower molar with dysodontiasis;Need to perform a mucoperiosteal flap;Need to perform osteotomy and/or odontotomy.The exclusion criteria were:Pregnancy;Breastfeeding;General contraindication to surgery.

After that, the participation in the clinical trial was proposed to the patient, together with a clarification on the various aspects and methods of the procedure. If the patient agreed to participate, he/she had to fulfill and sign the form of the informed consent that must be returned signed no later than the next visit.

Once the patient had accepted to participate in the study, the appointment for the surgical procedure was planned and the patient was asked to consent for the acquisition of clinical pictures and videos too. The whole flow chart of the randomized trial is reported in Figure 10, according to the Consort 2010 Diagram.

Surgical technique is the same used for extraction of the lower third molar. Patients enrolled in the study will not be exposed to any additional risk than the usual surgery techniques. In Figure 11 a sequence of snapshots from the performed interventions.

## 4. Results

For the 90 interventions we reported the average and the standard deviation of the trunk and neck rotation angles, according to different visual aids or the naked eye. We did not distinguish between left and right side interventions since rotation angles with respect to the sagittal plane show the same magnitude.

In Table 1, we summarized the trunk and neck rotation values of all interventions in the three groups obtained using the aforementioned quaternions.

### WMSD Risk Estimation Based on RULA Table

In this section we focus on the RULA evaluation of WMSD risk detailing the content of three scoring tables (named Table A, B and C in Figure 5) to provide evaluation of exposure risk to WMSDs. The assessment is based on the analysis of the arm, wrist, neck, trunk, and leg position.

The same surgeon in the same dental setting performed all the extractions of the third lower molars. In this way, we have the same postural conditions to compare the three groups of magnification accurately. According to the RULA worksheet the surgeon’s posture can be described as follows: The upper arms were located parallel to the trunk (Figure 5, Step 1). In the front view, the upper arms were abducted from the trunk. The shoulders were not raised and in our setting the arms are not supported. We could assign a score of 1 to the blue box of the final upper arm score.

The lower arms were placed between 60 and 100 degrees from the upper arms, and one arm worked repetitively across the midline. The final lower arm score is 2 in the pink box (Figure 5, Step 2).

The wrist is bent and twisted near the end of the range and we assigned, in Step 3, a value of 2 in the green box for the wrist twist score, while a value of 3 for the total wrist score (the yellow box) is value of 4. We can now merge all the results obtained and calculate the final score of Step 5 (result of Table A) that is 4 for all interventions. In Step 8, based on Table C, we then obtained the value of 5 (purple box) collecting the muscle and load score (Steps 6 and 7). The posture during chairside work is mainly static (see below for more details), and the repeated actions occur more than four times per minute (score 1 in Step 6), but none of the instruments weighed more than 4.4 lbs (score 0, Step 7). Steps 10, 11, 13 and 14 were also, in this case, the same for all the acquisitions.

The data collected from the markers showed a forward bending of the trunk between 3 and 12 degrees. The twist was from 2 to 5 and side bending from 3 to 9 degrees. Despite the wide variability of the data, we assigned a score of 4 (blue box of Step 10) for all interventions.

Legs and feet are supported, so we assigned a score of 1 in the light-yellow box of Step 11.

Steps 13 and 14 are similar to Steps 6 and 7 and, also in this case, a score of 1 was assigned to Step 13 for the repetitiveness of the actions and 0 to Step 14 for the low weight of loads.

The result for Step 15 from Table C depends mainly on Step 9, which represents the neck score and is the only variable that changes significantly in all interventions.

When considering Step 9 for the operating microscope, the forward bending (pitch) is less than 10° and twist and side bending are negligible, obtaining a value of 1, while for surgical loupes and naked eye we found a significant side bending and twist in both cases (even wider for the surgical loupes with respect to the naked eye) that, together with the forward bending, give a final value of 4.

In conclusion, we received a final value for Table C (in Figure 5) of 4 for the operating microscope and 6 for surgical loupes and the naked eye.

According to the final risk evaluation in Figure 12, we found a low risk for the operating microscope and a medium to high risk for surgical loupes and the naked eye.

An aspect that has little consideration in RULA analysis is long stillness in static postures, since it is uncommon in fatiguing activities; however, we believe that this feature is relevant, in particular for the operating microscope, where the same posture is kept for the whole intervention.

In Figure 13, we plotted the persistence of each static posture of the neck, according to the use of the naked eye, medical loupes, or surgical microscope. In particular, the pitch angle of the neck is represented against the persistence of the same posture in terms of seconds; the analysis is performed over all 90 acquisitions and the results are presented in terms of occurrences. The obtained histogram is then normalized to the total number of occurrences, giving the probability for each pitch angle and for each length of time the posture is kept. It can be seen that the pitch angle of the neck is larger for the naked eye with respect to surgical loupes or the operating microscope, according to Table 1, but the persistence is longer for the operating microscope (average value of 10 s) with respect to the other two approaches. The impact of prolonged staticity on MSD deserves further investigation; in RULA tables a static posture or a frequently repeated action (Step 13 of Figure 5) is penalized just as a ‘+1’ score, however, a more accurate analysis for the muscular stress requires the adoption of an electromyography device, but this is out of the scope of this article.

## 5. Conclusions

The goal of our study was to investigate if the use of magnification in dentistry could improve posture for dental practitioners and reduce the risk of onset of WMSDs. To accurately assess dentist posture, we decided to perform the same intervention on all patients, in the same settings, with the same operator. We proposed an advanced approach to evaluate the dentist posture accurately and objectively. A set of markers were applied on the back of a customized tight T-shirt worn by the dentist during the whole operation that was acquired using a commercial camera [32]. Thanks to this approach we were able to follow the 3D position and orientation of all the limbs involved in a specific activity during the job execution. Tracking the markers allowed us to know the posture of the operator and the time he/she kept an awkward position. The proposed approach removed subjective aspects related to the observer and his/her angles estimated from images. Furthermore, acquisitions were gained from a back view of the dentist and did not interfere with his/her activity.

In order to assess the feasibility of our experimental protocol, we chose to test it on the same surgeon in 90 interventions. In further studies, we aim to use the same protocol on a larger number of dentists introducing an electromyographic analysis to evaluate the correlation of the proposed method with each single muscle stress.

The final evaluation of the WMSD risk is performed using the RULA method, largely adopted for work-related ergonomics measures. Thanks to the use of three scoring tables, it provides a detailed evaluation of the exposure to risk factors focusing on upper arm, lower arm, wrist, neck, trunk, and legs.

Following a general procedure for entire body posture assessment [14], and considering a dentist in a sitting posture, we can see that in all interventions the lower limbs are in the same position. According to this consideration, we focused only on RULA (just for upper limbs) and not on its counterpart REBA (Rapid Entire Body Assessment) [33]. We adopted a standard set-up without a customized ergonomic stool and dental chair. The operator’s arms were not supported, and the backrest of the dental chair was not present.

Thanks to the proposed approach, every angle in the RULA table was evaluated accurately during the whole period of each intervention; we have also been able to evaluate the persistence of each static posture, offering the opportunity to evaluate its impact on muscular static stress as an ongoing activity.

In analyzing the different postures for different operating conditions (naked eye, medical loupes or surgical microscope), we found that the distance from the patient’s mouth was the most relevant variable involving different neck and trunk forward bending, and, in a smaller portion, neck and trunk twist and side bending.

Considering data of trunk and neck movements, we could see that the medical loupes gave slight improvements in terms or neck and trunk rotations with respect to the naked eye, while the surgical microscope induced significantly smaller rotations.

Thanks to the proposed method, we have been able to estimate RULA parameters in an objective way allowing a much more accurate and detailed analysis of the limb posture for fatiguing works. Our approach also lays the foundation to develop a new approach for limb posture assessment where accurate estimations, relative rotations, and static persistence are considered.

Considering the final RULA score, which provides the risk estimation for WMSD, we found that, for naked eye and medical loupes interventions there is a high risk, confirming the results in [5,6], while, for the surgical microscope, the total risk is classified just as ‘low’.

In conclusion, the proposed approach, when applied to dental surgery, allowed us to estimate the risk of WMSD for different visual aids, pointing out that using the surgical microscope is less fatiguing than medical loupes and the naked eye. In order to get more exhaustive statistics, we need to extend our study to a wide variety of dentists of different gender, age and expertise with the opportunity to correlate our results with proven WMSD stated by the surgeons. Furthermore, the proposed method allows for a cheap, minimally invasive, accurate and objective evaluation of working posture to define the correlation between upper limb posture and WMSD risk that can be easily extended to other caring professions.

## Figures and Tables

**Figure 1 sensors-21-07717-f001:**
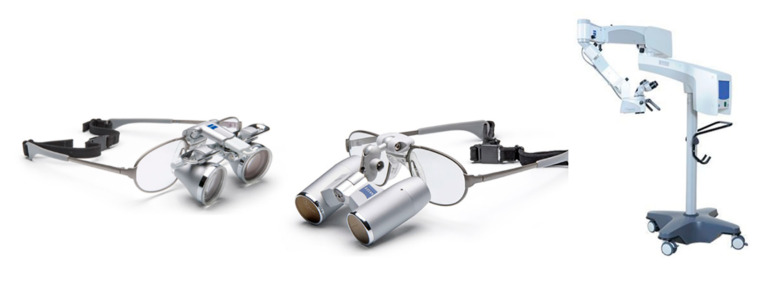
Galilean loupes (the ones used in the trial), Keplerian loupes (another widely adopted surgical magnification tool), and an operating microscope (Movena) Zeiss^®^, Jena, Germany.

**Figure 2 sensors-21-07717-f002:**
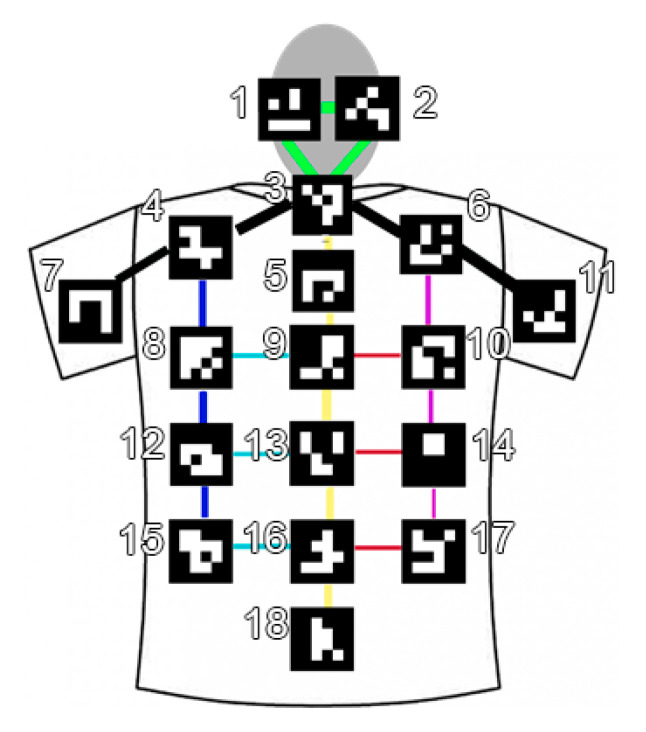
A schematic representation of the position of the 18 ArUco planar markers sewn on the T-shirt worn by the operator. The identification number is used in the following.

**Figure 3 sensors-21-07717-f003:**
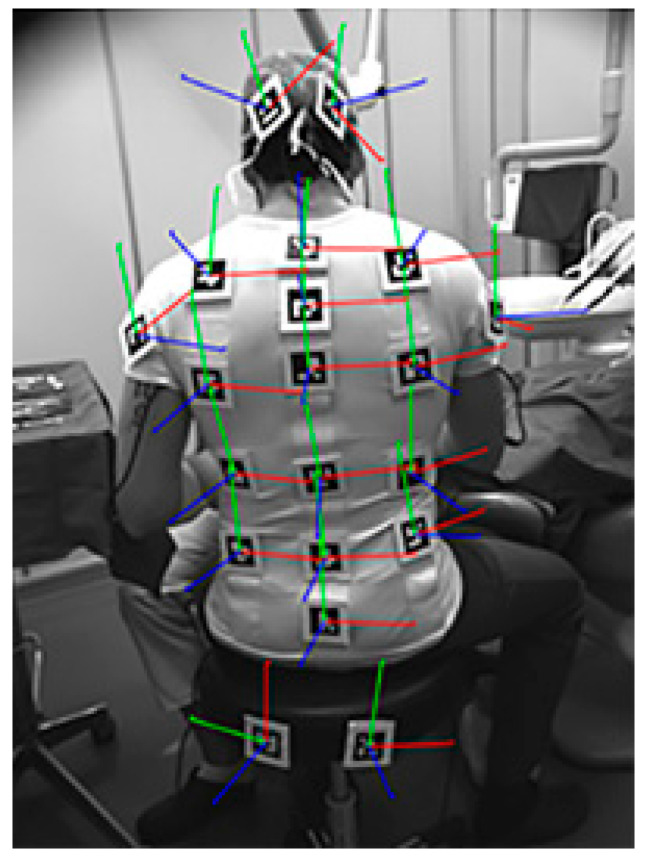
Superposition of the estimated axes for each marker in an acquired frame. The red, green, and blue vectors represent the x, y, and z axis, respectively.

**Figure 4 sensors-21-07717-f004:**
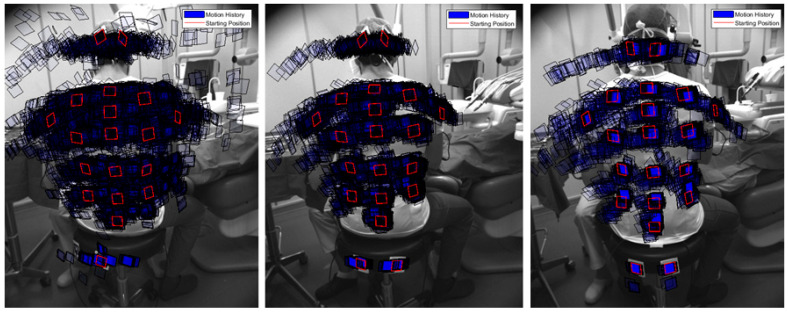
Superposition of the ArUco marker displacements detected during different surgical procedures starting from the reference position: (**Left**): An intervention based on the naked eye. (**Center**): An intervention using medical loupes. (**Right**): An intervention with the surgical microscope.

**Figure 5 sensors-21-07717-f005:**
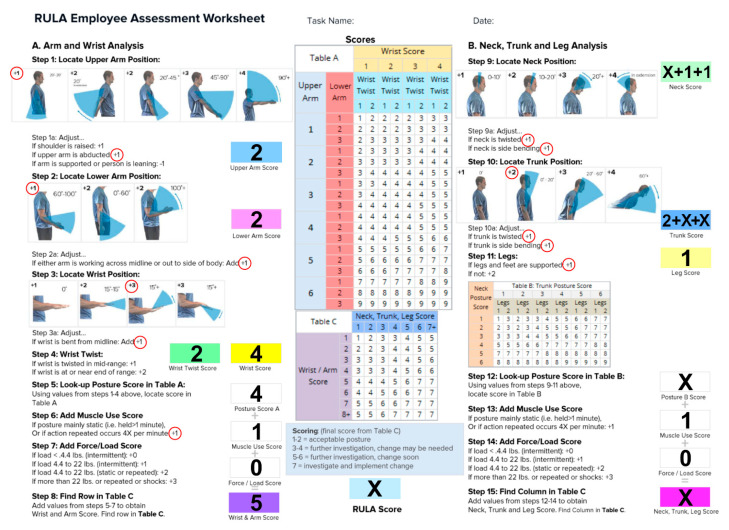
The filling sequence of a RULA form according to what is explained in the article (courtesy of ErgoPlus). The “X” represents values that, in our analysis, change according to the visual aid adopted. Further details in the text.

**Figure 6 sensors-21-07717-f006:**
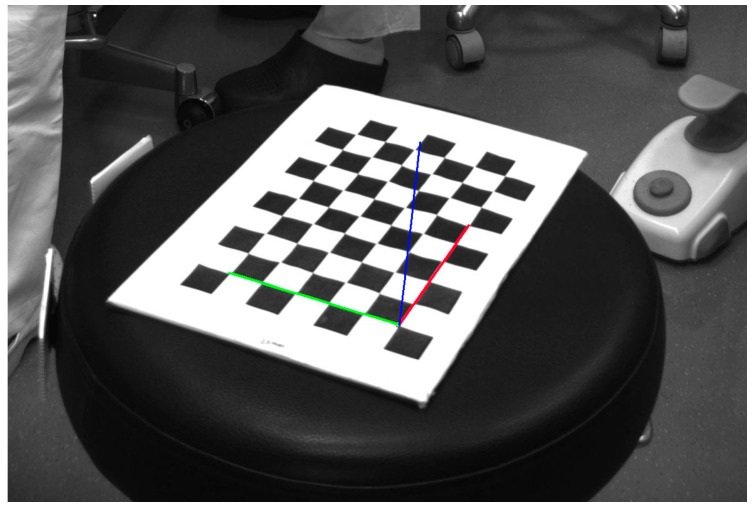
A checkerboard placed on a horizontal plane to get a global reference system; it is referenced as “ground”; using this set of axes we can consider the role of gravity in each limb posture.

**Figure 7 sensors-21-07717-f007:**
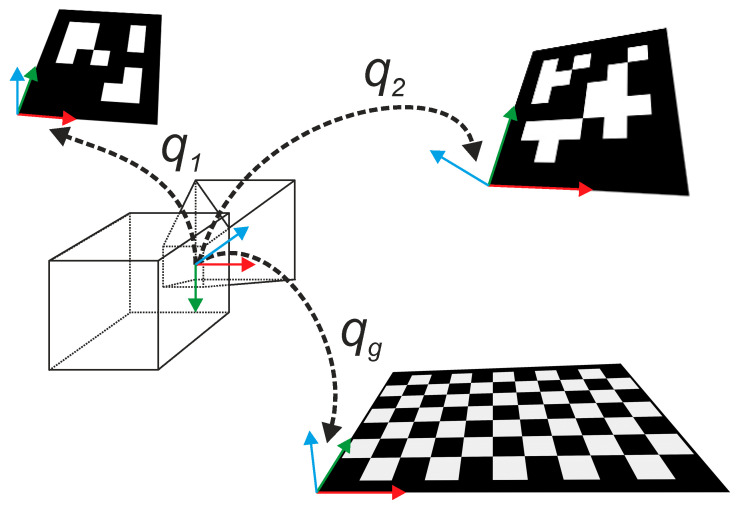
Using the ArUco module and the checkerboard calibration tool of OpenCV we get rotation matrices and translation vectors of each marker from the camera reference frame. Here we depict the quaternions **q**_1_ and **q**_2_ representing the rotation of two ArUco markers and the quaternion **q*_g_*** from the checkerboard representing the ground floor.

**Figure 8 sensors-21-07717-f008:**
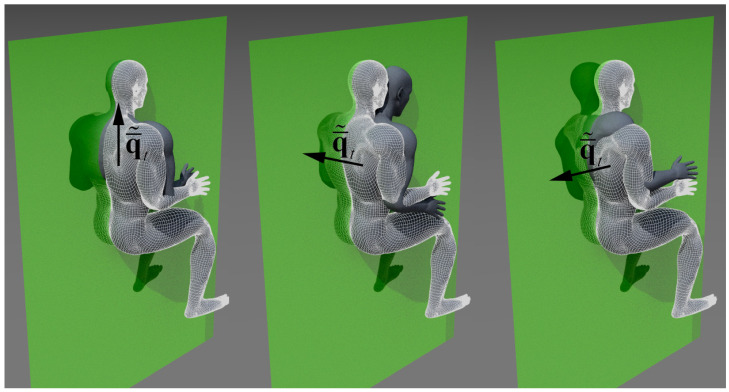
Average relative quaternion ${\tilde{\bar{q}}}_{t}$ for three different rotations. The green plane is the sagittal plane and the quaternions are represented just for their vector part. Rotations are relative to the reference posture represented with the checkered mesh: (**left**) yaw; (**center**) pitch, the quaternion is orthogonal to the sagittal plane; (**right**) roll, the quaternion is horizontal in the sagittal plane.

**Figure 9 sensors-21-07717-f009:**
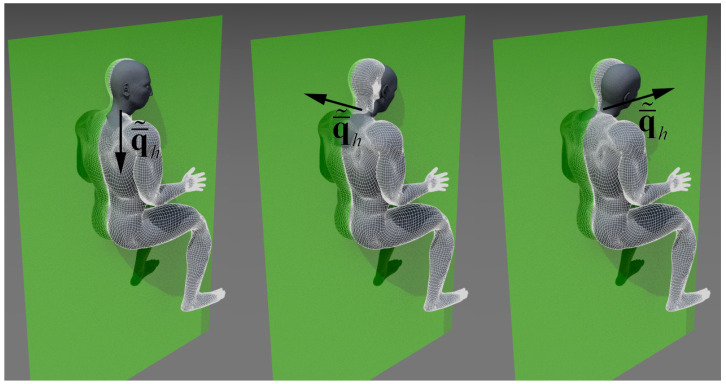
Similarly to Figure 8, the average relative quaternion ${\tilde{\bar{q}}}_{h}$ for the head-neck is represented. Yaw (**left**), pitch (**center**), and roll (**right**) relative rotations are represented accordingly.

**Figure 10 sensors-21-07717-f010:**
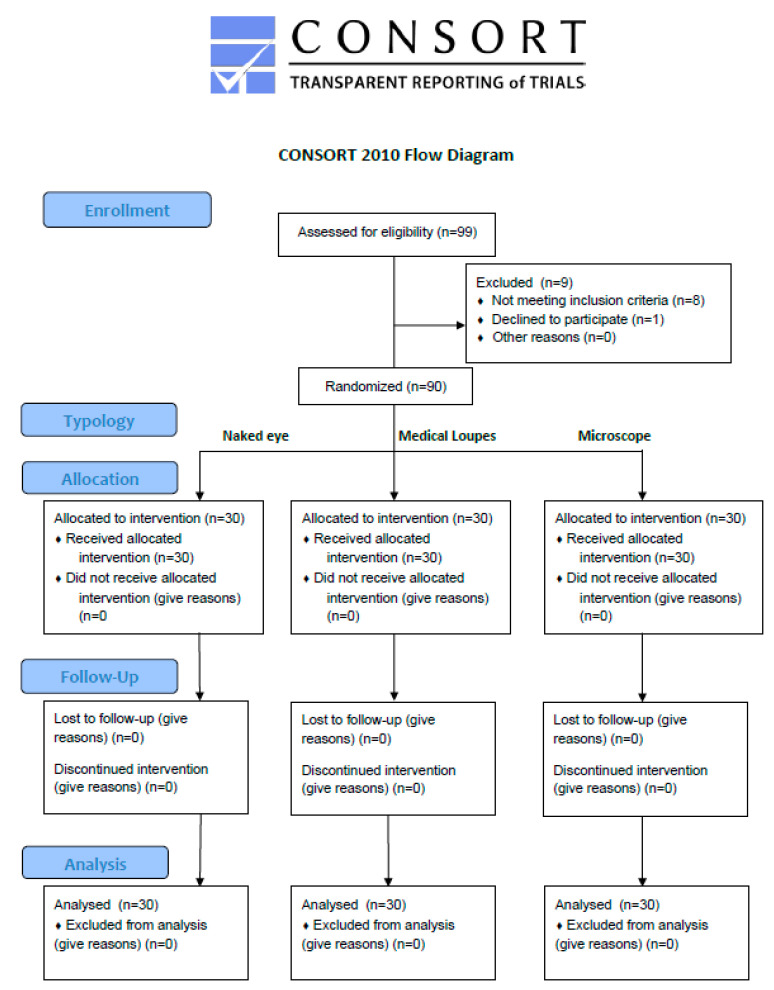
Consort 2010 Flow Diagram realized according to the CONSORT statement recommendations for randomized trials [31].

**Figure 11 sensors-21-07717-f011:**
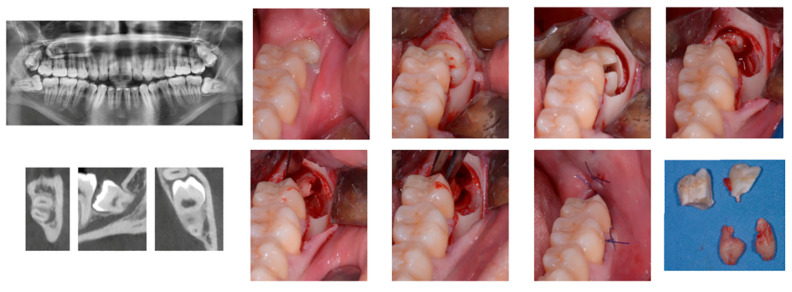
Access to the third molar was performed by a mucoperiosteal flap and a vestibular osteotomy with a fissure burr under continuous irrigation. If necessary, a crown and/or root sectioning was performed using the same fissure burr. After extraction the alveolus was inspected, curetted for granulation tissue removal, and irrigated with sterile saline solution. A 4/0 monofilament nylon suture was used to close the wound without tension. The choice of the third molars was done since their extraction represents a difficult and complex task to be carried out in the rear part of the mouth where large occlusions require, in many cases, the surgeon to keep uncomfortable positions for long periods. Due to the presence of many nerves in the operation zone, high attention must be kept for the whole intervention, increasing the stress and the muscular tightening of the practitioner.

**Figure 12 sensors-21-07717-f012:**
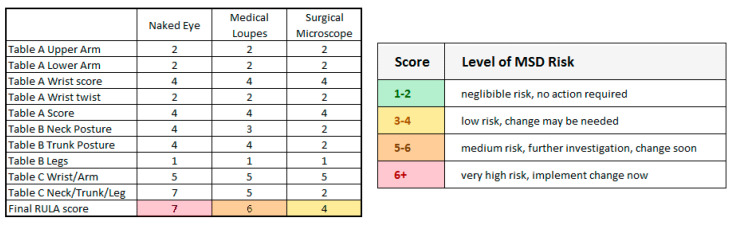
On the left, the different scores obtained for each body part according to the RULA table. On the right, we report the musculoskeletal disorders risk related to the final RULA Score from Table C in Figure 5.

**Figure 13 sensors-21-07717-f013:**
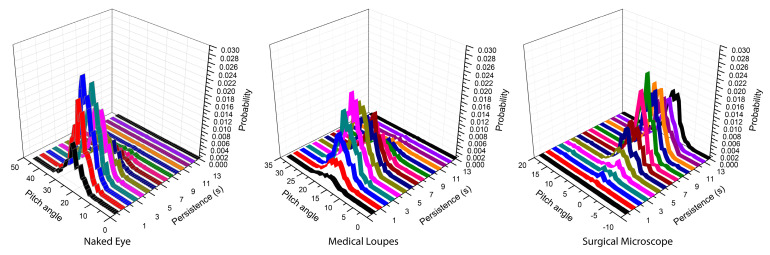
Analysis of static posture persistence of the neck pitch angle for naked eye, medical loupes and surgical microscope. The “persistence” represents the length of time the same pitch angle of the neck is kept during interventions. Each graph represents a normalized two-dimensional histogram of the occurrences; the vertical axis can then be assumed as the probability that a specific pitch angle is kept for a specific length of time.

**Table 1 sensors-21-07717-t001:** Rotation angles for neck and trunk: the first number is the average value in degrees while the second one inside parentheses is the standard deviation (also in degrees).

	Neck			Trunk		
	Forward Bending	Side Bending	Twist	Forward Bending	Side Bending	Twist
Operating Microscope	3.63 (1.02)	1.13 (0.52)	3.17 (1.69)	3.65 (1.77)	3.01 (1.31)	2.52 (0.60)
Surgical Loupes	15.67 (3.00)	11.73 (2.45)	17.20 (2.01)	11.25 (2.82)	8.92 (3.92)	5.58 (2.00)
Naked Eye	26.03 (3.94)	7.30 (2.31)	18.57 (5.51)	12.83 (4.31)	8.12 (2.91)	5.60 (3.23)

## Data Availability

No available data.

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
