# Peer review of "Posture Assessment in Dentistry for Different Visual Aids Using 2D Markers"

_sensors, 2021, doi:10.3390/s21227717_

Round 1

Reviewer 1 Report

Paper review Manuscript ID: sensors-1373528 Dear authors, Thanks for submitting a very interesting manuscript. The authors developed a new method on tracking the clinical surgeons’ postures using plate markers and used this proposed method to keep tracking the difference of using loupes, operating microscopes, or naked eyes to complete extractions. The manuscript is generally in good structure, but the reviewer has certain doubts that need to be clarified before considering for publication.

1. Some brief information was given in the introduction section at the beginning of the manuscript, however at the end of this section, the current research gap and what we are expecting to read in the following manuscript is not too clear. The reviewer felt a sudden stop at the end of this section.

2. It is great to see that the choose of method (the use of tools) was randomized. It seems that Figure 1 lists two loupes and one microscope, however in the line 117-119, it is clearly listed that you will have three comparisons, then the use of two different loupe is not clear to the reviewer. Please clarify.

3. The primary outcome only shown until line 140+. I believe it will make the manuscript clearer if you mention this expectation at the end of the introduction section.

4. Are the marks only placed on the back of the body? Any side view photos would support the understanding of your motion capture design.

5. A few questions in regard to the accuracy and application of this new proposed method:

• Since you are using the rotation matrices and translation vectors of each marker compared with reference frame to determine the posture and rotation, the initial set up of the marks are important, can you please illustrate the requirement of using and placing these markers? To be more specific, do you need to align the center of the marks in a way to start the data collection? Do you stick marks while the subject is sitting? Etc.

• Additionally, there are lots of existing and trustable motion capture systems and can accurately capture and motion of human body with minor errors. Why you need to propose a new method?

• How accurate it is? Any validation has been completed previously? The accuracy is very critical singe you are measuring the rotation and you only have variables in neck rotation and trunk inclination.

6. The reviewer recommends adding real photos along with the RULA figure as the example to show how you did the calculation and came up with the assessment. Additionally, is the measurement taken at one point in time based on one posture? Or a series of time upon the completion of the extraction. This information is important.

7. In the result section, please further explain the meaning of posture persistence in clinical field.

8. The manuscript is presented by using lots of small paragraphs, it would be better if it is more integrated.

Author Response

Reply to the reviewer can be found in the attached document.

Reviewer 2 Report

See attached

Author Response

(The authors gave the same response as above.)

Round 2

Reviewer 1 Report

The manuscript has been significantly improved.